# Detrainment and braking of snow avalanches interacting with forests

Louis Védrine[1,2], Xingyue Li[3], and Johan Gaume[1,4]

[1]School of Architecture, Civil and Environmental Engineering, Swiss Federal Institute of Technology, Lausanne, Switzerland
[2]Université Paris-Saclay, ENS Paris-Saclay, DER Génie Civil et Environnement, Gif-sur-Yvette, France
[3]Department of Geotechnical Engineering, College of Civil Engineering, Tongji University, Shanghai, China
[4]WSL Institute for Snow and Avalanche Research SLF, Davos, Switzerland

**Correspondence:** Xingyue Li (xingyueli@tongji.edu.cn)

**Abstract.** Mountain forests provide natural protection against avalanches. They can both prevent avalanche formation in release zones and reduce avalanche mobility in runout areas. Although the braking effect of forests has been previously explored through global statistical analyses on documented avalanches, little is known about the mechanism of snow detrainment in forests for small and medium avalanches. In this study, we investigate the detrainment and braking of snow avalanches in forested terrain, by performing three-dimensional simulations using the Material Point Method (MPM) and a large strain elastoplastic snow constitutive model based on Critical State Soil Mechanics. First, the snow internal friction is evaluated using existing field measurements based on the detrainment mass, showing the feasibility of the numerical framework and offering a reference case for further exploration of different snow types. Then, we systematically investigate the influence of snow properties and forest parameters on avalanche characteristics. Our results suggest that, for both the cold and warm snow parametrized in our simulations, the detrainment mass decreases with the square of the avalanche front velocity before it reaches a plateau value. Furthermore, the detrainment mass significantly depends on snow properties. It can be as much as ten times larger for warm snow compared to cold snow. By examining the effect of forest configurations, it is found that forest density and tree diameter have cubic and square relations with the detrainment mass, respectively. The outcomes of this study may contribute to the development of improved formulations of avalanche–forest interaction models in popular operational simulation tools and thus improve hazard assessment for alpine geophysical mass flows in forested terrain.

**KEYWORDS:** forest, avalanche, snow, detrainment, numerical modeling, material point method, gravitational mass movements.

## 1 Introduction

The expansion of human activity in mountains has increased the risk associated with snow avalanches (De Biagi et al., 2012) which threaten infrastructures and human lives. Forests can mitigate snow avalanche hazards without expensive construction of heavy protective measures like concrete dams. Hence, it is crucial to understand how forests affect the avalanche dynamic behaviour. A forest can have two protective effects against avalanches. It stabilises the snowpack in the release area (Viglietti et al., 2009) and reduces avalanche mobility in the runout zone. The second aspect has often been neglected because of the field

observation that large avalanches can destroy forests with no significant deceleration (Bartelt and Stöckli, 2001). However, it
has been shown that for small to medium avalanches (e.g. < 10000 m$^3$), forests are normally not destroyed and can notably
reduce the avalanche run-out distance (Teich et al., 2012a; Perzl et al., 2021).

Predominant numerical tools for modelling snow avalanches use two-dimensional approaches, like depth-averaged avalanche
dynamics model (Christen et al., 2010). This method is computationally efficient and a powerful tool for hazard mapping, but
cannot directly capture the interaction between individual trees and an avalanche. Therefore, many studies have worked on
modeling the effect of a forest indirectly. There are mainly two approaches to simulate the protective effect of a forest against
avalanches: the frictional approach and the detrainment approach (Feistl et al., 2012). The first accounts for the breakage of
trees and debris entrainment by increasing the friction parameters of the Voellmy law (Voellmy, 1955) compared to open
unforested terrain (Gruber and Bartelt, 2007; Takeuchi et al., 2018).The second approach is based on field observations and
has a more solid physical meaning than the friction model. The mass and momentum of the snow stopped behind the trees are
directly removed from the flow, which naturally leads to deceleration and run-out shortening of the avalanche. Assuming that no
mass is entrained in the forest area, the detrainment rate $\dot{Q}_\mathrm{d} = -\frac{1}{\rho}\dot{M}_d$ is added to the mass balance, with $M_d$ the detrainment
mass which corresponds to the mean mass stopped in the forest per unit of area. The rate of detrainment is quantified with
a detrainment coefficient $K$ (Feistl et al., 2014), which links the temporal derivative of the detrainment mass with $\mathbf{V}$, the
depth-averaged velocity of the avalanche as follows : $\frac{dM_d}{dt} = -\frac{K}{||\mathbf{V}||}$.

The two methods can both empirically recover the reduction in mass and momentum of an avalanche in an efficient way, but
may encounter difficulties and give unsatisfactory predictions in practical applications. According to Teich et al. (2012b), the
friction approach is not effective for modelling small and medium avalanches in forested terrain, as the run-out distance was
overestimated even by using the smallest turbulent friction coefficient. The detrainment approach highly relies on the data from
field avalanches, with which the detrainment coefficient $K$ can be determined to apply the detrainment function in numerical
modeling (Teich et al., 2014). Based on existing studies, both the forest type and the snow properties have a crucial effect on
the detrainment coefficient $K$ (Teich et al., 2012b, 2014; Brožová et al., 2020). However, there is no systematic investigation
on the controlling factors of the snow detrainment by forests under well-controlled conditions.

This study aims to quantify the amount of snow caught in forests and explore the mechanism of the detrainment, with
comprehensive consideration of different avalanche features and forest configurations. We use a three-dimensional Material
Point Method (3D MPM), by which the fractures, collisions, and large deformations involved in snow avalanches can be well
captured (Gaume et al., 2018, 2019; Li et al., 2020, 2021). More importantly, individual trees and their interactions with snow
avalanches can be explicitly modelled without relying on empirical laws. Compared to the popular two-dimensional tools for
modelling snow avalanches, the 3D MPM can fully resolve flow variations in all dimensions. Based on the simulations in
this study, we can better understand the forest-avalanche interaction, reveal laws governing the effect of the key influencing
factors on snow detrainment, and offer a basis for systematic calibration of the detrainment approach (Feistl et al., 2014) for
operational purposes.

## 2 Methodology and setup

### 2.1 Numerical method

#### 2.1.1 The material point method (MPM)

Our simulations are performed using the Material Point Method (MPM), which is a hybrid Eulerian–Lagrangian method suitable to deal with large strains (Stomakhin et al., 2013). In MPM, Lagrangian particles are used to carry the information of position, velocity and deformation gradient, while the Eulerian grid is used to compute the equations of motion and for updating the particle states. The governing equations are based on mass and momentum conservation as follows:

$$\frac{D\rho}{Dt} + \rho \nabla \boldsymbol{v} = 0, \tag{1}$$

$$\rho \frac{D\boldsymbol{v}}{Dt} = \nabla \boldsymbol{\sigma} + \rho \boldsymbol{g} = 0, \tag{2}$$

where $t$ is time, $\boldsymbol{v}$ is velocity, $\rho$ is density, $\boldsymbol{g}$ is the gravitational acceleration and $\boldsymbol{\sigma}$ is the Cauchy stress expressed by the elasto-plastic constitutive relation:

$$\boldsymbol{\sigma} = \frac{1}{J} \frac{\partial \Psi}{\partial \mathbf{F}_{\mathrm{E}}} \mathbf{F}_{\mathrm{E}}^{T} \tag{3}$$

where $\Psi$ is the elasto-plastic potential energy density, $\mathbf{F}_{\mathrm{E}}$ is the elastic part of the deformation gradient $\mathbf{F}$ and $J = \det(\mathbf{F})$.

In this study, the transfers between particles and grid use the Affine Particle-In-Cell (APIC) method (Jiang et al., 2015). Compared to the Particle-In-Cell (PIC) and Fluid Implicit Particle (FLIP) techniques, the APIC approach allows both to stably remove the numerical dissipation and to preserve angular momentum in addition to linear momentum.

#### 2.1.2 Large-strain elastoplastic model

For modeling snow, we use the modified Cam Clay model with associative flow rule developed by Gaume et al. (2018). Given a stress $\boldsymbol{\tau}$, a mean effective stress and a deviatoric stress can be respectively obtained as $p = -\frac{1}{3}\mathrm{tr}(\boldsymbol{\tau})$ and $\boldsymbol{s} = \boldsymbol{\tau} + p\mathbf{I}$, where $\mathbf{I}$ is the identity matrix. The Mises equivalent stress can then be derived as $q = \sqrt{\frac{3}{2}\boldsymbol{s} : \boldsymbol{s}}$. The yield surface in the space of the $q$–$p$ is defined as:

$$y(p,q) = (1 + 2\beta)q^2 + M^2(p + \beta p_0)(p - p_0) \tag{4}$$

where $\beta$ is the ratio between tensile and compressive strength and represents the cohesion, $p_0$ is the consolidation pressure and $M$ is the slope of the cohesionless critical state line, which denotes the internal friction of snow. For $y(p,q) \leq 0$, the material follows the Hooke's law (St Venant-Kirchhoff Hencky strain), otherwise it behaves plastically. The hardening and softening of the material are modelled by respectively expanding and shrinking the yield surface according to the law:

$$p_0 = \frac{E}{3(1 - 2\nu)} \sinh\left(\zeta \max(-\epsilon_{\mathrm{v}}^{\mathrm{p}}, 0)\right) \tag{5}$$

where $\zeta$ is the hardening factor, $E$ the Young's modulus, $\nu$ the Poisson's ratio and $\epsilon_{\mathrm{v}}^{\mathrm{P}} = \log(\det(\mathbf{F}^P))$ is the volumetric plastic deformation.

## 2.2 Model setup

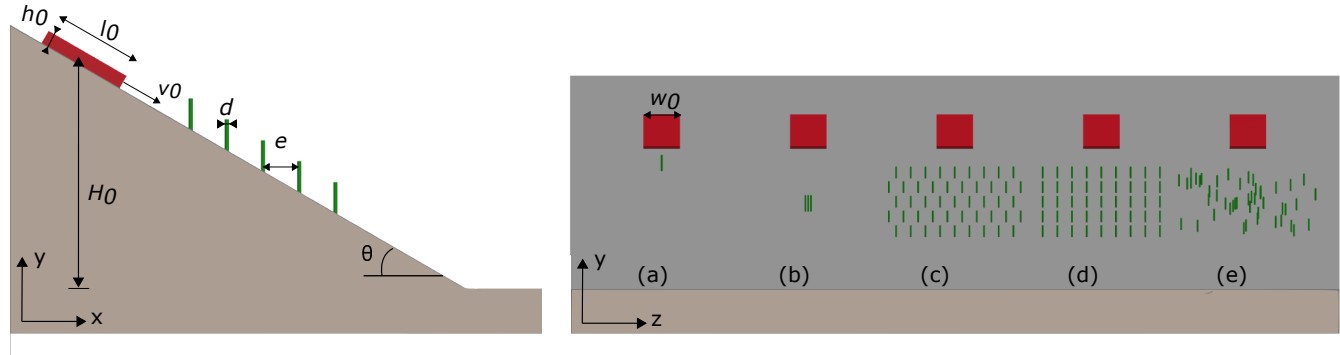

**Figure 1.** Model setup of MPM simulations of snow avalanches impacting to trees. The left and right are side and front views respectively. Five arrangements of trees in the right include: (a) one tree; (b) three threes; (c) regularly staggered trees; (d) regularly aligned trees; (e) random trees.

Different simplified set-ups are designed to test the protective role of different forest structures with various types of snow.
As shown in the left of Fig. 1, the terrain is composed of an ideal slope with a constant inclination of $\theta = 30°$ and a horizontal deposition zone. The bed friction coefficient of the terrain is assumed to be 0.5. The snow sample is initially placed at the top of the slope at a height of $H_0 = 50.6$ m with a prescribed initial velocity of $\boldsymbol{v}_0$. We define $h_0 = 1$ m, $l_0 = 20$ m and $w_0 = 20$ m as respectively the height, length, and width of the released snow. For all the snow types simulated in this study, they share the same Young's modulus ($E = 3$ MPa), Poisson's ratio ($\nu = 0.3$) and initial density ($\rho_0 = 200$ kg/m$^3$) (Li et al., 2020). As
the material point method is a continuous approach, each material point corresponds to a piece of snow, therefore the snow density is the bulk density, including air and snow. Five types of tree arrangement are designed as shown in the right of Fig. 1, including (a) one tree; (b) three threes with an identical diameter and at the same elevation; (c) a regularly staggered forest composed of trees in a staggered arrangement; (d) a regularly aligned forest composed of trees in an aligned arrangement; (e) a random forest in which trees are randomly located. All the trees are initially at the downstream of the released snow. In the
setup of c)&d)&e), the width of the forests (along z) is larger than the maximum avalanche width, and the length of the forests along the flow direction is 40 m.

In addition to the different tree arrangements, the forest density $\rho_{\mathrm{forest}}$ (trees/ha) and the tree diameter $d$ are also varied to study their effect on snow detrainment. The forest density in this study is defined as the number of trees per hectare. In the case of a regularly staggered forest, the forest density is controlled by the spacing between the trees $e$ (Fig. 1). In this study, we

assume that the trees are unbreakable and the trunk surface is rough. Therefore, the trees are modeled as rigid obstacles with a no-slip boundary condition. The height of the trees is set to 10 meters so that the avalanches do not overtop the trees.

In order to both well model the trees and shorten the computation time, we use a Eulerian grid size of $dx = 0.15$ m if it is not specified and 8 particles per grid. The time step is constrained by the Courant-Friedrichs-Lewy (CFL) condition and the elastic wave speed to guarantee the simulation stability. The simulation data are exported every $1/12$ s.

## 3 Calibration with a documented case

To calibrate the friction of snow and verify other parameters (e.g. snow density, Young's modulus, mesh size) adopted in our numerical modelling, we simulate the snow detrainment due to a group of 3 trees according to the field observation in Feistl et al. (2014). This case with 3 trees is selected for the calibration as the setup is simple and the field measurement data are available for comparison. Based on the real condition (Feistl et al., 2014), the diameter of trees is set to 1 m and the spacing between trees is 0.33 m in our simulation. The simulation results are quantitatively compared with the field data in terms of the volume and height of the deposited mass (i.e. detrainment) behind the trees.

By assuming the shape of the detrainment mass as a wedge, Feistl et al. (2014) calculated the detrainment volume as

$$W = \frac{lh_{\mathrm{w}}^2}{2\,tan(\theta)} \tag{6}$$

with $l$ the width of the group of trees, $h_{\mathrm{w}}$ the wedge height, $\theta$ the slope angle. In our simulations, the detrainment volume is obtained by summarising the volume of all the particles in the detrainment mass without the assumption of the wedge shape. The height of the detrainment mass is calculated as the vertical distance from its free surface to the tree foot (as illustrated in Fig. 2). As summarised in Table 1, five simulation cases with different snow friction coefficient $M$ were conducted, while

**Table 1.** Deposition height and volume of snow in simulations with different M and in the field observation

| $M$ | $W(m^3)$ | $h_w(m)$ |
|---|---|---|
| 1.5 | 29.9 | 3 |
| 1.2 | 28.7 | 2.5 |
| 1 | 27.2 | 2.7 |
| 0.9 | 25.7 | 2.5 |
| 0.8 | 14.3 | 2.1 |
| Goal (Observation) | 13.85 | 2 |

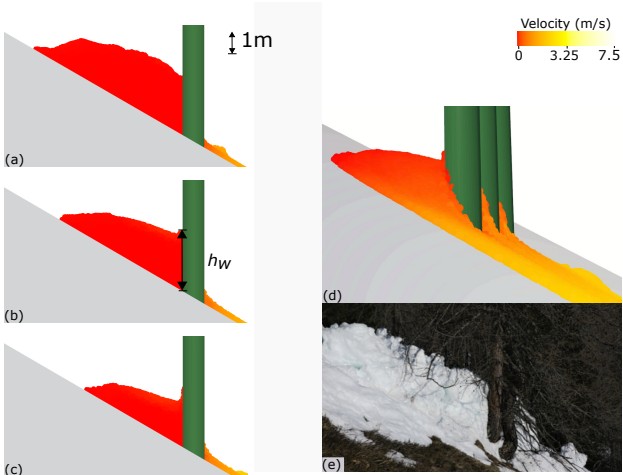

**Figure 2.** Cross-sectional view (passing through the center of the middle tree) of the snow stopped behind a group of three trees for different values of $M$: (a) $M$=1.2, (b) $M$=1, (c) $M$=0.8; perspective view of the stopped mass (d) in the simulation with $M = 0.8$ and $d = 1$ m and (e) from the field observation (Feistl et al., 2014)

all other parameters are fixed (e.g. the tension compression ratio $\beta = 0.3$, the hardening factor $\zeta = 1$, the initial consolidation
pressure $p_0^{ini}$ = 30 kPa). With the reduction of snow friction from 1.5 to 0.8, the volume and height of the detrainment mass decreases and the case with $M = 0.8$ has a good agreement with the field data. In addition, Fig. 2 shows that the shape of the detrainment mass does not differ much with the change of snow friction. The simulation with $M = 0.8$ gives a similar profile of the detrainment mass as that observed from the field (Fig. 2d&e). This calibration case demonstrates the good feasibility of our numerical tool in capturing the key features of detrainment and serves as a reference case for the study on the effect of
snow properties in the following section.

## 4   Results

A qualitative study of the forest-avalanche interaction will be first presented where different avalanche characteristics will be introduced (subsection 4.1). Then, the main forest factors influencing the detrainment mass in the case of a regular staggered forest will be discussed (subsection 4.2) and a newly proposed unique law for estimation of the detrainment mass will be
suggested (subsection 4.3). Finally, we will show an energy investigation involving the braking process as well as the influence of snow properties and forest structure (subsection 4.4).

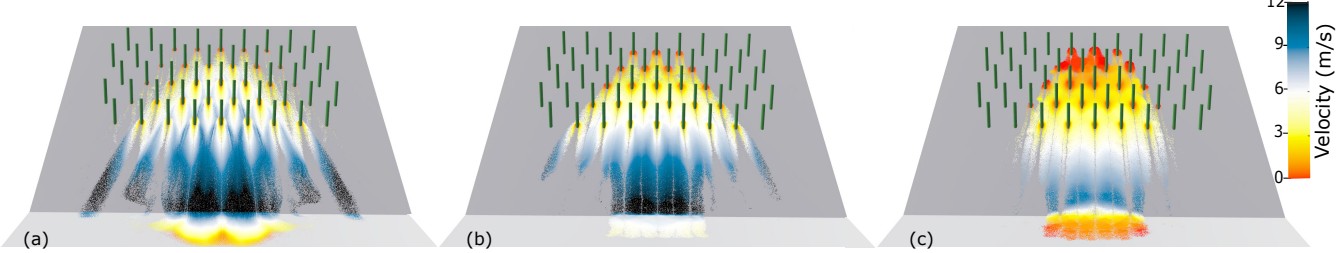

**Figure 3.** Flow profile for 3 different flow regimes: (a) Case 1, (b) Case 2, (c) Case 3, for a regular staggered forest at $t = 10$ s, $e = 8$ m, $d = 1$ m, $v_0 = 10$ m/s

**Table 2.** Snow properties adopted in the MPM modeling of the flows for three typical flow regimes

| Case id | $M$ | $\beta$ | $\zeta$ | $p_0^{\mathrm{ini}}$(kPa) |
|---------|-----|---------|---------|---------------------------|
| Case 1  | 0.5 | 0       | 1       | 3                         |
| Case 2  | 0.8 | 0.3     | 1       | 30                        |
| Case 3  | 1.2 | 0.3     | 1       | 30                        |

## 4.1 Influence of avalanche features

### 4.1.1 Snow properties at the forest scale

In this study, different snow properties are adopted in the MPM modeling, to study their influence on the braking effect of
the forest. The snow properties are defined by the parameters of the Modified Cam Clay model (Gaume et al., 2018): the friction coefficient $M$, the tension compression ratio $\beta$, the hardening factor $\zeta$ and the initial consolidation pressure $p_0^{\mathrm{ini}}$. These parameters have their physical basis and can be determined according to physical properties of snow and/or parametric study (Li et al., 2020). In particular $\beta p_0^{\mathrm{ini}}$ denotes the isotropic tensile strength, $M$ is linked to the the internal friction angle of snow $\phi$ as follows $\phi = \sin^{-1}(\frac{3M}{6+M})$ (Sadrekarimi and Olson, 2011), and the hardening factor reflects how fast the load
increases with the displacement in the plastic stage. According to our systematic study on the effect of snow properties on the avalanche behavior (Li et al., 2020), it was found that the tensile strength $\beta p_0^{\mathrm{ini}}$ and $\beta M$ consistently increase from cold to warm avalanches, which suggests that these two terms control the different snow behaviors.

   Based on the calibration case reported in section 3, $\beta p_0^{ini} = 9$ kPa, $\beta M = 0.24$ (Case 2 in Table 2), we can obtain different snow types by changing $\beta p_0^{ini}$ and $\beta M$. Firstly, to reproduce the behavior of a cold snow (Case 1 in Table 2), we decrease the
tensile strength $\beta p_0^{\mathrm{ini}} = 0$ kPa and $\beta M = 0$, which leads to a cohesionless and low friction snow. And to capture the behavior

of a wet and warm snow, $\beta p_0^{\text{ini}} = 9$ kPa and $\beta M = 0.36$ (Case 3 in Table 2) are adopted, giving a relatively cohesive and frictional snow with granules and blocks.

As illustrated in Fig. 3, a regular staggered forest is used for the three cases in Table 2. It is apparent that the snow properties have a significant effect on the accumulated snow behind the trees. Indeed, the total amount of stopped snow increases with the

internal friction and the cohesion of snow. According to our simulations, some arches due to jamming effect (Feistl et al., 2014) appear in Case 3 especially at the upstream of the avalanche. This occurrence of the arches promotes the stabilization of snow on the slope and is tightly related to the high internal friction and cohesion of snow. The jamming effect and the formation of the arches observed in the avalanche-forest interaction are similar to the behavior of a granular flow in a two-dimensional hopper (Lai et al., 2001).

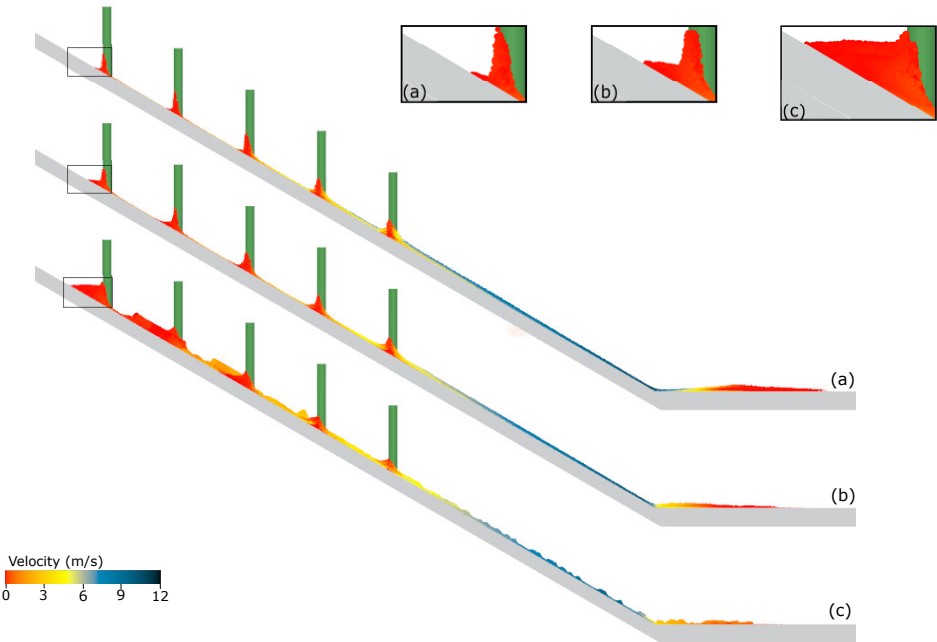

**Figure 4.** Profile of accumulated snow for different flow regimes: (a) Case 1, (b) Case 2, (c) Case 3, regular staggered forest at $t = 10$ s, $e = 8$ m, $d = 1$ m, $v_0 = 10$ m/s

Independent of the snow properties, there are a lot of defections as the flow front is split into many branches (Fig. 3), in contrast to the flow without forest obtained from our simulation. Fig. 3 demonstrates that the more cohesionless the snow is, the more the snow is laterally spread in the forest and gets around the trees easily. As shown by Luong et al. (2020), for the cohesionless flow regime (Case 1), the snow on the side of the flow moves very fast. After being initially deflected, this lateral snow moves in a diagonal channel without collision, even if a part of this effect is due to the arrangement of trees like a Galton

board.

In addition to the lateral motion of the avalanches, the length and height of the stopped snow are further analysed with the side view as illustrated in Fig. 4. The insets at the upper corner of Fig. 4 show the stopped mass (in red) at the uppermost trees.

It is interesting to observe that the height of the wedge of stopped mass does not vary much with the different snow properties. Meanwhile, the length of the snow wedge in the flow direction changes notably. With the more cohesive and frictional snow from Case 1 to Case 3, the length of the wedge increases and its free surface becomes more horizontal (inset c). According to Fig. 4, the snow properties also affect the evolution of stopped mass along the flow direction. While the shapes of the snow wedges appear to be similar along the slope in Cases 1&2, the profile of the snow changes significantly in Case 3. As blocks and granules exist in Case 3, there is more snow stopped by the trees at the upstream. It is noticed that the uppermost trees in Case 3 have less stopped mass than the second uppermost trees, mainly due to the severe initial impact of the avalanche on the uppermost trees.

### 4.1.2 Avalanche velocity

To define the detrainment mass stopped by the trees, a criteria based on the evolution of particle velocity is used. If a snow particle has a velocity smaller than 0.5 m/s throughout the flowing process, it is considered as stopped and detrained. This definition borrows the concept by Feistl et al. (2014), where the snow is directly removed from the avalanche mass once it is stopped.

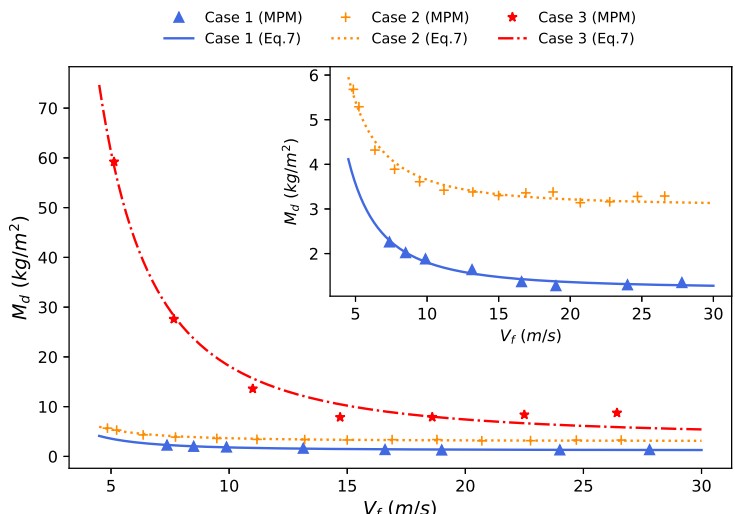

**Figure 5.** Evolution of the detrainment mass per unit of area of snow with the velocity (regular staggered forest, $e = 8$ m)

The effect of avalanche velocity on detrainment snow is studied by imposing an initial velocity $v_0$ ranging from 0 to 24 m/s. Figure 5 shows the evolution of the detrainment mass with the front velocity at the entrance of the forest for the different snow properties (in subsection 4.1.1). The detainment mass per unit area $M_d$, is calculated by dividing the total detrainment mass by the total area of influence. The latter is obtained by multiplying the number of trees with snow accumulation by the area of influence of each tree ($e^2$).

Firstly, the evolution of the stopped snow with snow properties is quantified in Fig.5, a very strong effect is observed, for low speeds, between the Case 1 and 3, the total amount of snow stopped is multiplied by 10. However, for each type of snow the same behaviour is observed, the detrainment mass decreases when the velocity increases and tends to a constant value for high velocities. The initial negative correlation with the low velocities is in agreement with the field observation that a lower velocity leads to more stopped snow (Feistl et al., 2014). Therefore, we propose that the detrainment mass decreases with velocity square according to

$$M_d(V_{\mathrm{f}}) = \frac{p_1 + p_2 \, V_{\mathrm{f}}^2}{V_{\mathrm{f}}^2} \tag{7}$$

with $V_{\mathrm{f}} = \sqrt{v_{\mathrm{xf}}^2 + v_{\mathrm{yf}}^2 + v_{\mathrm{zf}}^2}$ the front velocity of the avalanche when it enters in the forest. The front velocity is calculated based on the velocity of the first $15\%$ of the particle flow to eliminate the effect of scattering particles at the avalanche front. The parameters of this model ($p_1(\mathrm{kg/s^2})$, $p_2(\mathrm{kg/m^2})$) (Fig. 5) are identified based on least squares method (Tab. 3) and both depend on the snow properties. $p_2$ represents the detrainment mass reached for very high velocities (Eq. 7), it is the mass stopped independently of the velocity.

Similar to the effect of velocity on the detrainment mass, Kyburz et al. (2020) showed that the impact pressure increases with the velocity square. This increase of the impact pressure causes the breakage of snow during its collision with the trees, which leads to a lower volume of snow stuck behind the trees. The detrainment mass for the highest velocity is rather due to the type of snow, this part of snow is initially compacted against the trunk at the front of the avalanche and is not impacted by the velocity.

## 4.2 Influence of forest parameters

### 4.2.1 Tree diameter

The influence of tree diameter is studied for the one tree configuration with a mesh size of $0.1$ m for a better modeling of thin trees. In the case of a single tree, we use the mass stopped $m_d$ (kg), not normalize with the area around the tree as the area is difficult to define. For all the snow properties, both the maximum and the final masses stopped $m_d$, have the same evolution trend with the tree diameter (Fig. 6). Feistl et al. (2014) proposed the following cubic model for the detrainment volume $W$ induced by single-tree interaction:

$$W = \frac{d^3}{12 \tan(\theta) \tan(\delta/2)^2} \tag{8}$$

with $d$ the diameter of the tree, $\theta$ the slope angle, $\delta$ the top wedge angle of the pyramid formed by stopped snow. Comparing the simulated final mass with the previous analytical proposal of Feistl et al. (2014), we can notice that this model is close to the flow in Case 2 calibrated for a tree diameter of 1 meter, it proves that the calibration is consistent. However, the numerical result shows that the order 3 model in Eq. 8 is not verified. This observation is in accordance with the analysis of the shape of the snow wedge behind the tree, indeed the height of the wedge is not linearly influenced by the tree diameter, whereas the

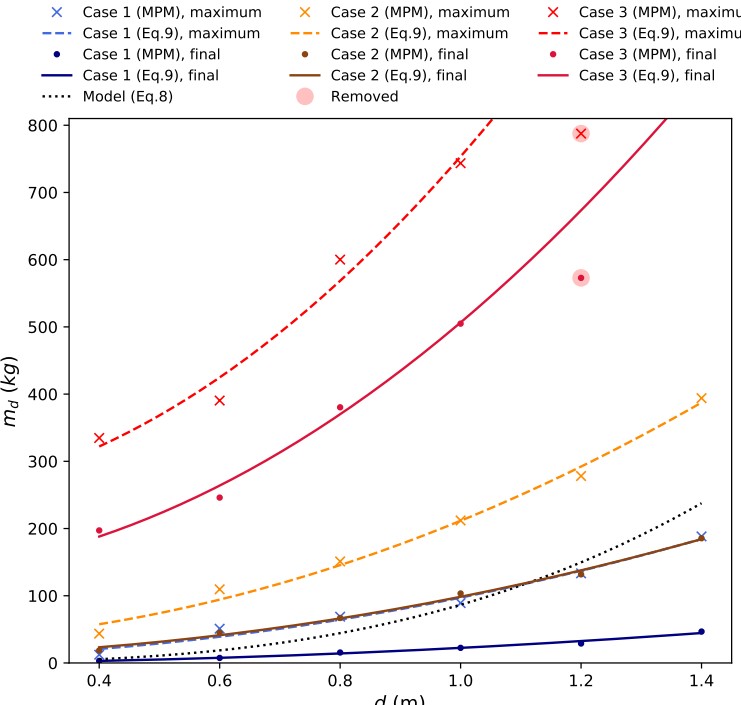

**Figure 6.** Evolution of the detrainment mass observed for the one tree arrangement and the fits with the tree diameter for 3 flow regimes : Case 1, Case 2 and Case 3 with respectively a front velocity of $12.5\,\mathrm{m/s}$, $10.9\,\mathrm{m/s}$ and $10.75\,\mathrm{m/s}$, 'maximum' refer to the maximum mass stored behind the tree and 'final' refer to the final mass stored. A slope of $30°$, and a top wedge angle of $60°$ (from measurements Feistl et al. (2014)) are used for the theoretical model (Eq. 8).The removed point denotes a special case not considered in the proposed square relation in Eq. 9, since the entire avalanche in this case is stopped due to the low flow velocity and the high tree diameter. Please note that it is a coincidence that Case 1 maximum and Case 2 final agree well

diameter has a large effect on the length and the width of the wedge. Thus, the evolution of the stopped mass with the tree diameter follows a square law for each type of snow.

While we dispose of the evolution of the detrainment mass with the tree diameter for the one tree arrangement, in the case of an arrangement of several trees, the spatial distribution of the detrainment mass shows that trees interact with each other (Fig. 7). Comparing to the one tree result, more snow is stopped for the first row than for the one tree case, and for the lower rows the detrainment mass can be either underestimated or overestimated. This is not only a geometric effect, because the pattern of the spatial distribution of detrainment mass behind each tree depends on the velocity (Fig. 7). Thus, it reveals the difficulty to predict the distribution of detrainment mass. For a better understanding of the phenomenon, the study must be carried out at the forest scale, and the study of the forest density would help to consider the collective behavior and interaction of the trees.

For the study of a forest, the influence of tree diameter is examined by keeping the same forest density and changing the basal surface area per hectare $\Phi = \frac{A_{\mathrm{trunk}}}{A_{\mathrm{forest}}}$,where $A_{\mathrm{trunk}}$ is the total area occupied by the trunks and $A_{\mathrm{forest}}$ the total forest area.

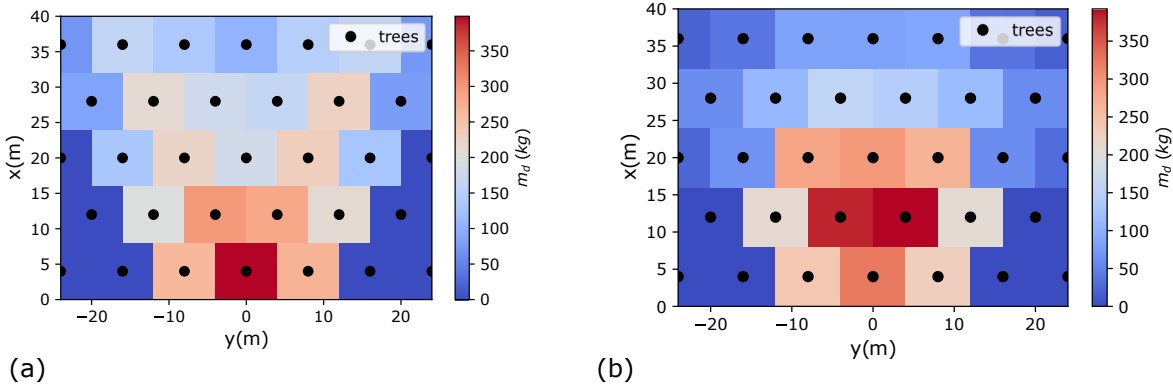

(a)

(b)

**Figure 7.** Distribution of the maximum detrainment mass behind each tree during the avalanche (regular staggered forest, $d = 1$m, $e = 5.5$ m, snow properties: Case 2), (a) $v_0 = 10$ m/s, (b) $v_0 = 22$ m/s

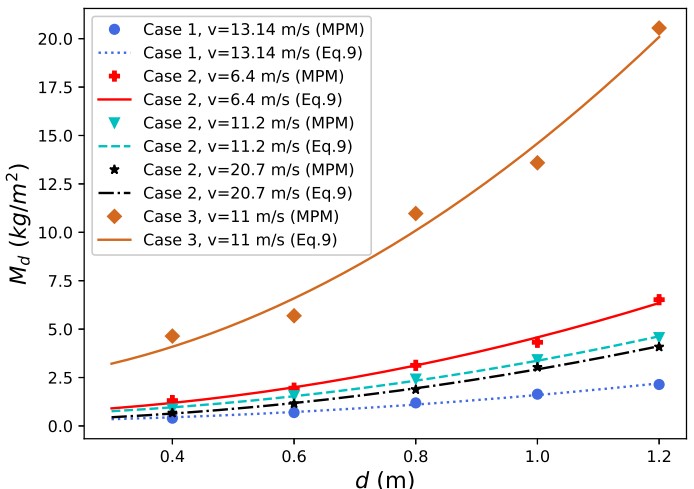

**Figure 8.** Evolution of the detrainment mass per unit of area with the tree diameter for different types of snow and front velocity (regular staggered forest, $e = 8$ m)

Figure 8 depicts that the same behaviour is observed for different velocities and snow properties, and the trend is similar to the case of one tree. These results suggest that the detrainment mass evolves according to a square law with the tree diameter (Eq.(9)), with $p_3$ and $p_4 (1/\text{m}^2)$ evaluated based on least squares method (Tab. 3).

230 $\quad M_d(V_f, d) = M_d(V_f) \left( p_3 + p_4 \, d^2 \right)$ $\hfill$ (9)

#### 4.2.2 Forest density

The effect of forest density is studied with a constant tree diameter of $0.6$ m in Fig. 9a and a fixed Stand Density Index (SDI) (Reineke, 1933) in Fig. 9b, respectively. The range of the adopted SDI from $400$ to $1347$ trees/ha in Fig. 9a corresponds to normal to very dense forests (Abegg et al., 2020). Figure 9a shows the evolution of the detrainment mass under the effect of the forest density, for different front velocities. In previous models (Feistl, 2015), it was assumed that the increase of the forest density only increases the number of trees/obstacles and consequently the number of wedges, which led to a linear law between the mass stopped and the forest density. However, our results suggest that the forest density has an influence on the stopped mass behind each individual tree. Indeed, compared to a forest with a lower density, a denser forest leads to more detrainment not only because there are more trees but also due to more stopped mass for each tree as snow jamming occurs. This increased detrainment due to the collective behavior of the trees is observed for all the different velocities simulated. Based on the obtained simulation data, the third power law is the best fitting model, therefore the proposed law in Eq. 9 can be improved with consideration of the effect of forest density accoring to Eq. 10 below

$$M_d(V_f, d, \rho_{\text{forest}}) = M_d(V_f, d)\left(p_5 + p_6\ \rho_{\text{forest}}^3\right) \tag{10}$$

with $p_5$ and $p_6 (\text{ha}^3/\text{trees}^3)$ evaluated based on least squares method (Tab. 3).

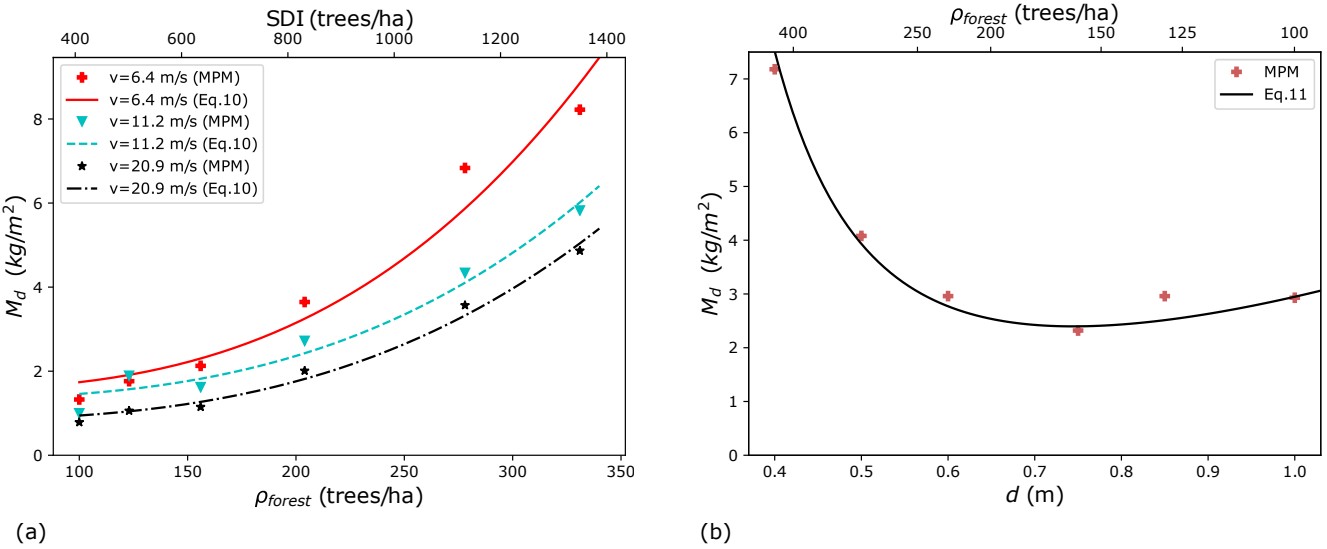

**Figure 9.** (a) Evolution of the detrainment mass per unit of area with the forest density for different front velocities (regular staggered forest, snow properties: Case 2, $d = 0.6$ m), (b) Evolution of the detrainment mass per unit of area with the tree diameter for a constant Stand Density Index of $925$ trees/ha (regular staggered forest, $v_0 = 10$ m/s, snow properties: Case 2)

The evolution of the detrainment mass with the tree diameter is shown in Fig. 9b, where the Stand Density Index is fixed at $925$ trees/ha and the forest density is changing. When the tree diameter is small ($d < 0.6$ m), the detrainment mass decreases

with the tree diameter, due to the reduction of the forest density and the number of trees. Indeed, less trees give a weaker collective effect of the forest, and thus stop less snow. A different trend is observed when the tree diameter becomes larger ($d > 0.6$ m), indicating the competing roles played by the collective effect of the trees and the individual tree effect. The larger the tree diameter, the more the stopped mass behind each individual trees, which changes the decreasing trend when the tree diameter is small. In addition to the simulation results, the prediction of the detrainment mass with the proposed model in Eq. 10 is shown in Fig.9b. As observed, the model prediction agrees well with the simulation data, showing the good performance of the model and the possibility to use the Stand Density Index in the model.

## 4.3   Model for detrainment mass

Based on the systematic study of the influence of the avalanche features and forest parameters, a model for the detrainment mass per unit of area can be proposed for a regular staggered forest (Eq. 11).

**Table 3.** Parameter value of the law (11) for different snow properties (Table 2)

| Type of snow | $p_1(\text{kg/s}^2)$ | $p_2(\text{kg/m}^2)$ | $p_3$ | $p_4(1/\text{m}^2)$ | $p_5$ | $p_6(\text{ha}^3/\text{trees}^3)$ |
|---|---|---|---|---|---|---|
| Case 1 | 58.7 | 1.2 | 0.12 | 0.84 | $7.81 \times 10^{-1}$ | $9.4 \times 10^{-8}$ |
| Case 2 | 58 | 3.1 | 0.12 | 0.84 | $7.81 \times 10^{-1}$ | $9.4 \times 10^{-8}$ |
| Case 3 | 1434 | 3.8 | 0.12 | 0.84 | $7.81 \times 10^{-1}$ | $9.4 \times 10^{-8}$ |

$$M_d(V_f, d, \rho_{\text{forest}}) = \frac{p_1 + p_2\, V_f^2}{V_f^2}\, \left(p_3 + p_4\, d^2\right)\, \left(p_5 + p_6\, \rho_{\text{forest}}^3\right) \tag{11}$$

with $V_f$ the frontal velocity of the avalanche when it reaches the forest, $\rho_{\text{forest}}(\text{trees/ha})$ the number of trees per hectare, $d$ the tree diameter, and $p_{[1,2,3,4,5,6]}$ the model parameters given in Table 3.

Figure 10 depicts the evolution of the detrainment mass predicted from Eq. 11 and that from MPM simulations, showing that the developed law in Eq. 11 predicts well the detrainment mass. As expected Case 1 and Case 3 have the smallest and largest detrainment mass respectively.

## 4.4   Energy analysis of the avalanche–forest interaction

### 4.4.1   Evolution of energy and energy dissipation

After the evaluation of the detrainment mass, it is needed to quantify the affect of the detrainment process in the braking effect of forest against avalanches. To understand the physical process of the braking effect, the analysis of the evolution of the kinetic and potential energy with and without forest is made (Fig. 11). As expected, we observe when the avalanche enters

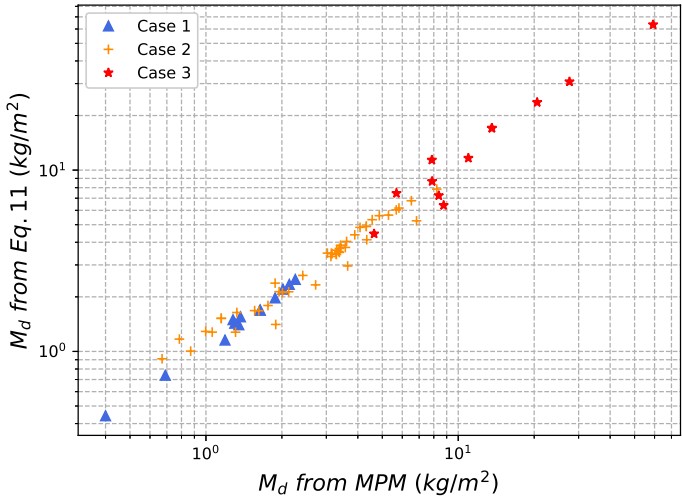

**Figure 10.** Evolution of the detrainment mass predicted with the model (Eq. 11) and with the observation, the coefficient of determination $r^2$ for the model prediction on the detrainment mass is $0.9912$

the forest that the kinetic energy decreases while without forest the kinetic energy continues to increase steadily. An important difference of kinetic energy between the simulation with and without forest when the avalanche front reaches the bottom of the slope is observed, which confirms the significant braking effect of forest. In addition, in term of potential energy, whereas the potential energy without forest decreases linearly as a function of time, with forest, due to the mass which stays on the slope, the potential energy decreases more slowly with time. Consequently at the end of the simulation, the potential energy does not vanish due to the detrainment mass stopped by the forest.

To understand the kinetic energy loss with the forest, two physical processes are considered, namely, detrainment and dissipation. First, a detrainment behavior accounts for the stopped snow mass behind the trees, whose potential energy is not converted to kinetic energy. Considering the avalanche potential energy with and without forest $E_{pf}$ and $E_{p0}$ respectively, the change of the potential energy due to the detrainment with the presence of the forest $E_{pd}$ can be defined as

$$E_{pd} = E_{pf} - E_{p0}. \tag{12}$$

Secondly, the dissipation process is associated with avalanche-tree interactions including collision and friction, which lead to plastic dissipation and random fluctuation. This dissipation due to the trees is defined as $\tilde{E}_f$, whose calculation is detailed in Appendix A.

Figure 12 shows the energy change due to the detrainment and dissipation processes (i.e. $E_{pd}$ and $\tilde{E}_f$), which contribute to the reduction of the avalanche kinetic energy with the forest. Taking the avalanche with forest as a reference, when its front enters the forest, the energy loss due to detrainment and dissipation start to grow (at around 2.3 s in Fig. 12). The increasing rate of $\tilde{E}_f$ tends to decrease with time, as the velocity of the flow decreases (see the kinetic energy in Fig. 11) and the dissipation by

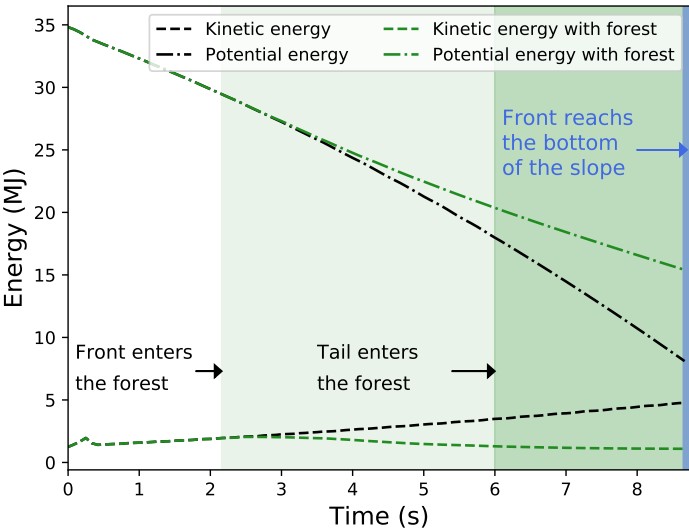

**Figure 11.** Temporal evolution of the kinetic and potential energy without forest and with a regular staggered forest (Case 2, $v_0 = 6$ m/s)

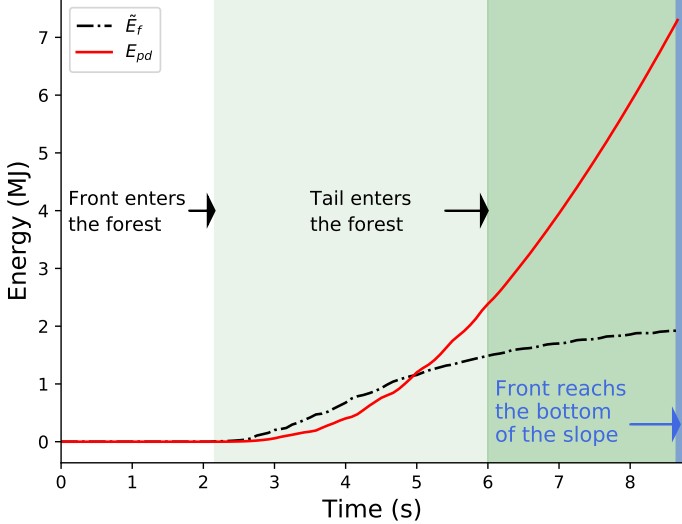

**Figure 12.** Temporal evolution of the detrainment energy and dissipation due to the forest (Case 2, $v_0 = 6$ m/s)

ground friction reduces. In contrast, the growth rate of $E_{pd}$ increases with time. Indeed, after small wedges are formed behind and around the trees, the spacing between trees decreases, which leads to faster blockage of incoming snow particles. It should

be noted that the forest enhances energy loss due to detrainment and dissipation, but meanwhile reduces the avalanche velocity. Hence, the presence of the forest leads to lower energy dissipation due to ground friction, as it is positively correlated with the avalanche velocity.

### 4.4.2 Influence of the forest structure and the snow properties on the energy loss

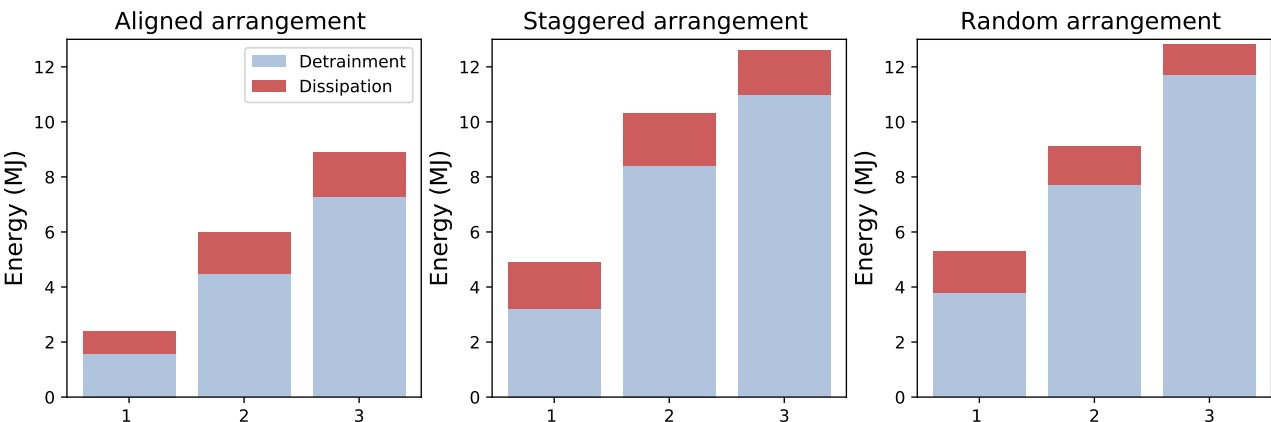

**Figure 13.** Energy detrainment and dissipation for different snow properties (1: Case 1, 2: Case 2, 3: Case 3 with $M$=1) and for 3 types of forest structure (regular aligned, regular staggered and random)

The same experiment is carried out for three types of snow (Table 2) and three types of forest (regular aligned, regular staggered and random) with the same forest density and tree diameter, in order to analyse the separate processes of detrainment and dissipation. Figure 13 depicts the detrainment and dissipation energies when the front of the avalanche reaches the bottom of the slope, where the front is defined as the first $1\%$ of the avalanche. It reveals that the detrainment represents a larger part of the energy loss compared to the dissipation. The detrainment increases with the cohesion and the internal friction of snow, whereas the dissipation due to forest changes slightly with the snow properties. For the three types of snow, the dissipation is higher for the regular staggered forest, due to the Galton board arrangement for which collisions and deflections are more frequent. The most notable difference for the different forest arrangements in Fig. 13 is that much lower detrainment is observed for the aligned forest, indicating that much less snow is stopped. Indeed, when trees are regularly aligned, a large part of the flow passes through the space between trees without collisions (Fig. 14). These results suggest that in the case of a regular forest, the staggered arrangement should be privileged for a better protective effect. The detrainment energy when the avalanche reaches the bottom of the slope is similar between the random and the regular staggered arrangements.

Given the same snow properties (e.g. Case 2), the final stopped mass with the aligned, staggered, and random arrangements is 2147 kg, 3002 kg, and 13061 kg respectively. The same trend is observed for other considered snow properties in this study (Cases 1&3). Compared to the regular aligned forest, the regular staggered forest stops more snow due to a smaller spacing (along the diagonal direction) between the trees (Fig. 14a&b). The random arrangement gives the highest stopped mass, as

each tree cluster serves as a dam and collectively block a large amount of snow (Fig. 14c). Consequently, in term of the final run-out distance, the avalanche in the random forest travels a shorter distance than that in the regular staggered forest, which suggests that the random one has a higher protective effect.

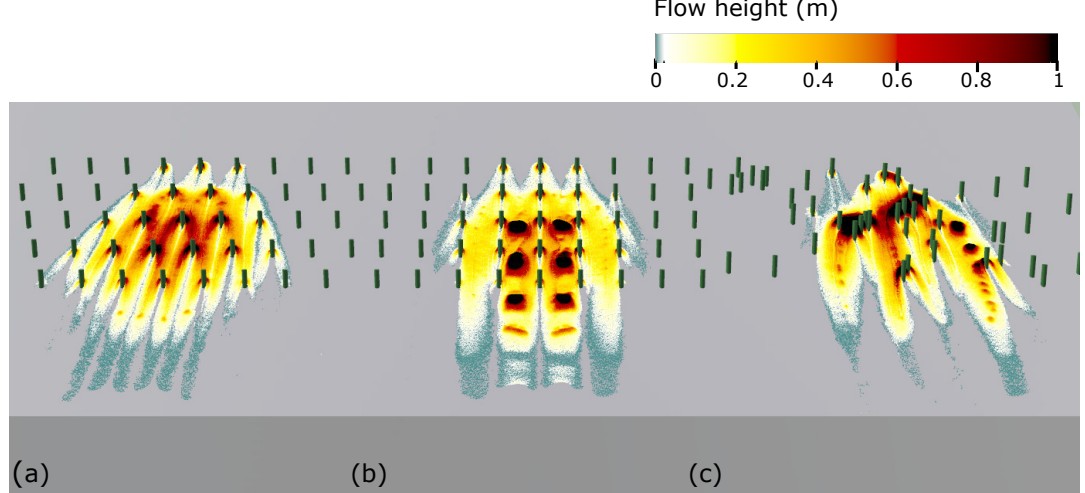

**Figure 14.** Flow height of the avalanche in a forest with (a) staggered arrangement, (b) aligned arrangement, (c) random arrangement, (Case 2, $v_0 = 6$ m/s)

## 5   Discussion and conclusions

In this study, we have explored the avalanche–forest interactions with the material point method (MPM) and an elastoplastic constitutive model. The aim is to highlight the dependencies of the detrainment mass and understand the physical process of the braking effect of forest with respect to avalanches dynamics. We have systematically examined the effect of snow properties, by studying both cold and warm snow. It is found that snow friction and cohesion have a large effect on detrainment mass behind the trees. Low snow friction and cohesion give fluid-like behaviour and highly sheared flows, and consequently little snow is stopped. When snow becomes more frictional and cohesive, large wedge and arches appear behind the trees, which contribute to large amount of snow stopped. Due to the scarcity of snow triaxial test (Scapozza and Bartelt, 2003), the snow parameters in our study are determined and calibrated according to their physical basis and previously conducted parametric study, nevertheless in the future, these parameters could be evaluated based on extensive triaxial tests for different snow types.

Interestingly, our analysis of the effect of avalanche velocity reveals that the detrainment mass decreases with the velocity and reaches a plateau value for high velocities (typically higher than $15$ m/s). While the detrainment mass linearly decreases with increasing velocity in Feistl et al. (2014), our numerical results suggest a decrease with velocity square. Knowing the relation between the detrainment mass per unit of area $M_\mathrm{d}$ and the detrainment coefficient $K = \frac{M_\mathrm{d} \cdot V_\mathrm{f}^2 \cdot cos(\theta)}{l_\mathrm{avalanche}}$ (Feistl, 2015) the detrainment rate can be assessed, with $V_\mathrm{f}$ the front velocity, $\theta$ the slope angle, $l_\mathrm{avalanche}$ the avalanche length. For an open

forest or old forests, Feistl et al. (2014) proposed a value of $K = 10$ Pa independently of the snow properties and the front velocity. With our simulation results, for the case 1, we obtain a value of $K = 5.31$ for a velocity of 7.4 m/s and $K = 32.4$ for 24 m/s, while for case 2, we obtain $K = 10.1$ for a velocity of 7.7 m/s and $K = 86.6$ for a velocity of 24.7 m/s. This underlines a high dependency of the detrainment coefficient on the snow properties and on the front velocity of the avalanche. The comparison with the previous value (Feistl et al., 2014) of $K$ for an open forest suggests that it is only valid for low velocities. When the avalanche velocity is high, the plateau stage of the detrainment mass obtained from our study is higher than the decreasing stage by (Feistl et al., 2014). Consequently, an implementation of our proposed velocity-based model would lead to higher detrainment mass, which results in lower runout distance and smaller velocity.

In addition, we have examined different forest arrangements, by varying the forest density and the tree diameter, which both can easily be identified from a forest inventory or remote sensing (Bebi et al., 2016). It is found that the detrainment mass increases with the cube of forest density and the square of tree diameter. Contrary to the geometric model proposed by Feistl et al. (2014), we found that the height of the wedge does not vary linearly with the tree diameter. This comprehensive numerical investigation allowed us to propose a unified law for the detrainment mass (Eq. 11). Therefore, it enables an estimation of the $K$ value as show before, which may serve as an input to 2D depth-averaged methods to better predict the detrainment mass.

From an energy point of view, our results suggest that the detrainment has the largest influence on the energy loss, which claims the use of the detrainment approach. However, the additional dissipation due to forest ($\tilde{E}_f$) is not negligible, although that the detrainment approach removes not only the potential energy of the removed particles but also their kinetic energy. In view of the kinetic energy levels compared to the dissipation added with forest, the kinetic energy of the particles removed seems too low to represent the dissipation added with forest. This result implies that an increase of the bed friction in combination to the detrainment approach could be justified to take into account the increase in energy dissipation linked to the effect of forest.

The presented research focuses on examining the interaction between forest and avalanches with idealised conditions, which enable to give new outlooks for further studies. Here, trees are modeled as rigid obstacles and with a no-slip boundary condition. Hence, this study is only valid for small to medium avalanches where the forest is not destroyed, and the trees act as obstacles. Secondly taking into account the true friction coefficient of the tree would allow, as suggested by Teich et al. (2014), to define a law for different species. In the future, adding a parameter to account for forest type, crown cover and surface roughness based on remote sensing data (Brožová et al., 2020) could help the operation of the proposed model and improve the evaluation of detrainment in Bayesian networks for risk assessment (Stritih, 2021).

Futhermore, slope angle can affect the detrainment mass, as it can significantly vary with the front velocity $V_f$. However, its effect on the geometry of the wedges is closely related other factors like avalanche velocity and bed friction, and therefore is difficult to be quantified independently. Future studies on the relation between the effect of slope angle and that of other factors will need to be conducted to introduce slope angle in our proposed model for the detrainment mass. In addition, the consideration of a slope with a constant inclination is an ideal condition. This could be further changed to any other shape to be more realistic and other tree arrangements could be explored as well. It would also be interesting to study longer forest with larger released volumes, to obtain longer permanent regimes, this would however significantly increase the computational time.

Moreover, this study focuses only on the avalanche dynamics and not on the release in forest area, therefore the conclusions on the effect of forest arrangement should be carefully interpreted in practical forest design for avalanche mitigation. As the avalanche release could be greatly affected by the forest as well, it is interesting to investigate the stabilization of snowpack under the effect of trees using MPM simulations in the future.

Despite the assumptions and idealisation applied in this study, it highlights the main factors influencing the detrainment mass in an unique law depending on snow mechanical properties, front velocity, and forest parameters. This outcome of this work can be applied not only for calibration of depth-averaged models used operationally (e.g., Christen et al. (2010)), but also for forest management. Moreover, according to energy analysis, a good calibration of the detrainment approach seems to be not enough to model all processes driving avalanche-forest interaction. This suggests that an approach combining detrainment and

friction increase could be more appropriate. Finally, although this study focused on snow avalanches, the methods used and the general relationships found are also relevant to other geophysical mass flows interacting with forests and can pose strong impacts on hazard assessment and risk management.

*Code and data availability.* All the data used in the figures in this paper are on Zenodo at https://doi.org/10.5281/zenodo.6121811. A detailed description of the MPM model can be found in a previous publication at https://www.nature.com/articles/s41467-018-05181-w.

*Video supplement.* Videos of the simulations presented in Figs. 2, 3, 4 and 14 are available on Zenodo at https://doi.org/10.5281/zenodo.5547386.

## Appendix A: Calculation of energy dissipation due to forest

By defining $\Delta e$ the additional dissipation at each time step, it is possible to separate it into two parts (Li et al., 2020): the dissipation inside the flow $\text{w}_{\text{int}}$ and through the boundary bed $\text{w}_{\text{b}}$, and in the case with forest, the dissipation $\tilde{\text{e}}_{\text{f}}$ is added. Hence, the total dissipation in the case without forest is $\Delta \text{e}_0 = \text{w}_{\text{b}_0} + \text{w}_{\text{int}_0} \approx \text{w}_{\text{b}_0}$, as the internal dissipation can be negligible

compared to the bed friction dissipation (Li et al., 2020). Similarly, in the forest case $\Delta \text{e}_{\text{f}} = \text{w}_{\text{b}_{\text{f}}} + \text{w}_{\text{int}_{\text{f}}} + \tilde{\text{e}}_{\text{f}} \approx \text{w}_{\text{b}_{\text{f}}} + \tilde{\text{e}}_{\text{f}}$.

As the velocity of the avalanche is reduced with forest, the frictional dissipation with the ground decreases, to evaluate the dissipation added with forest, we can not simply substract the dissipation obtained with and without forest. Therefore we use the equation A1a which allows us to calculate $\tilde{\text{E}}_{\text{f}}$ based on our simulations. Indeed, by replacing the expression of $\Delta \text{e}_0$ and $\Delta \text{e}_{\text{f}}$ (Eq. A1b), and by assuming that the bed dissipation $\text{w}_{\text{b}}$ depends linearly on the velocity (i.e $\frac{\text{w}_{\text{b}_{\text{f}}}(i.\Delta t)}{v_f(i.\Delta t)} \approx \frac{\text{w}_{\text{b}_0}(i.\Delta t)}{v_0(i.\Delta t)}$) (Eq.

A1b), the definition of the dissipation added due to forest is obtained (Eq. A1c). So at the time step $n$, with $\Delta t$ the time step, it is possible to obtain the total dissipation added by the forest $\tilde{\text{E}}_{\text{f}}$.

$$\sum_{i=0}^{n-1} \left( \frac{\Delta \text{e}_{\text{f}}(i.\Delta t)}{v_f(i.\Delta t)} - \frac{\Delta \text{e}_0(i.\Delta t)}{v_0(i.\Delta t)} \right).v_f(i.\Delta t) \tag{A1a}$$

$$= \sum_{i=0}^{n-1} (\frac{\text{w}_{\text{b}_{\text{f}}}(i.\Delta t) + \tilde{\text{e}}_{\text{f}}(i.\Delta t)}{v_f(i.\Delta t)} - \frac{\text{w}_{\text{b}_0}(i.\Delta t)}{v_0(i.\Delta t)}).v_f(i.\Delta t) \tag{A1b}$$

$$\approx \sum_{i=0}^{n-1} \tilde{\text{e}}_{\text{f}}(i.\Delta t) \tag{A1c}$$

$$= \tilde{\text{E}}_{\text{f}}(n.\Delta t) \tag{A1d}$$

*Author contributions.* L.V. performed the simulations, made the figures and videos, analyzed the results and wrote the paper under the supervision of X.L. and J.G. X.L. prepared the numerical setup of the simulations. J.G. designed the study and obtained the funding. All authors were involved in the writing of the manuscript and physical interpretation of the results.

*Competing interests.* The authors declare that they have no conflict of interest.

*Acknowledgements.* We acknowledge Dr. Perry Bartelt and Dr. Peter Bebi from SLF, Davos for insightful discussions on the topic of avalanche - forest interaction and the detrainment approach. We also thank Jean-Louis Gay from the Vaud Canton, Clara Streule and Francesc Molné from EPFL for their preliminary contribution on the topic in the framework of an EPFL Design Project motivated by a forest fire in an avalanche-prone area (Les Diablerets). Finally, we thank Dr. Martin Proksch from the Wallis canton and Andreas Tegethoff from Geoformer for constructive suggestions for the practical application of this work. Johan Gaume acknowledges funding from the Swiss National Science Foundation (Eccellenza project: grant number PCEFP2_181227; SPARK project grant number CRSK-2_195914).

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
