# Peer review of "Detrainment and braking of snow avalanches interacting with forests"

_Natural Hazards and Earth System Sciences, 2021_

## Author Comment (AC1)

**Response to Reviewer #1's comment on "Detrainment and braking of snow avalanches interacting with forests"**

Louis Védrine, Xingyue Li*, and Johan Gaume
*Correspondence: xingyueli@tongji.edu.cn
January 26th, 2022

We want to thank Referee #1 for his or her valuable comments and constructive suggestions that helped us to improve the quality of our paper. In the following, we provide detailed point-by-point answers to the comments raised by the reviewer.

The authors present an approach to quantify the effect of forest on a (slow) moving avalanche. To this end, they study the detrainment and braking due to trees by using a 3d model approach based on the material point method and a rheology previously proposed by one of the author.

They study varying forest stand composition and derive an empirical formula for practical use. The newly proposed formula is compared to one proposed by Feistl (2015).

The paper presents an important step to describe and quantify the interaction between avalanches and the efficiency of a forest to mitigate moving avalanches.

Reply: We thank the reviewer for this positive evaluation of our paper.

General comments:

The authors refer to the approach by Feistl (2015). At this point it would be valuable for the reader to get some more information about this approach without to have to look the full thesis.

Reply: The detrainment model proposed by Feistl (2015) aims to be implemented in a depth-average model like RAMMS (Christen et al., 2010). To reproduce the effect of forest with this model, the mass and momentum of the snow stopped behind the trees are directly removed from the flow. To do this, assuming that no mass is entrained in the forest area, the detrainment rate $\dot{Q}_d = -\frac{1}{\rho}\dot{M}_d$ is added to the mass balance, with $M_d$ the detrainment mass which corresponds to the mean mass stopped in the forest per unit of area. The rate of detrainment is quantified with a detrainment coefficient K (Feistl et al., 2014), which links the temporal derivative of the detrainment mass with **V**, the depth-averaged velocity of the avalanche as follows: $\dot{M}_d = -\frac{K}{\|\boldsymbol{V}\|}$.

Following the reviewer's suggestion, the clarification above has been added to the revised manuscript.

To enhance the practical usefulness, it would be nice, if the parameter study would cover more typical parameter combinations of forest stands. Firstly a tree diameter of 1 m refers to a rather very mature forest. Secondly, the stand density index (Reineke, 1933) in their example (figure 9) covers a range from SDI = [900, 3600], whereby the later sound rather high. Using a combination of 1 m diameter and 400 trees per hectare suggests an efficiency that a natural forest probably doesn't fulfill.

Reply: We thank the reviewer for pointing out the importance of using the Stand Density Index to enhance the practical usefulness of this study. Following the reviewer's suggestion, we have conducted new simulations using a smaller tree diameter (0.6 m) and a stand density index between 400 and 1347 tree /ha, which corresponds to a normal to a very dense forest (Abegg et al., 2020). The newly obtained results are shown as Fig. R1, which demonstrates similar

trends as the original Fig. 9 in the manuscript. We have replaced the original Fig. 9 with the figure below.

[Figure]

Figure R1. Evolution of the detrainment mass per unit of area with the forest density for different front velocities (regular staggered forest, snow properties: Case 2, $d = 0.6$ m).

The SDI would, properly, serve also better as fixed value for the comparison in figure 8 than the forest density.

Reply: Thanks for the suggestion. Initially, we chose to study the tree diameter at a fixed forest density because this allows us to obtain a simple and easily identifiable law from Fig. 8, in contrast to the case of a fixed stand density index. Following the reviewer's comment, we have added a new figure showing the effect of tree diameter with a fixed SDI as demonstrated in Fig. R2. This new plot allows us to check the predictability of the proposed model (e.g. Eq. 11 in the manuscript) and demonstrates that it is possible to use the model with the Stand Density Index parameter in Eq. 11.

[Figure]

Figure R2. Evolution of the detrainment mass per unit of area with the tree diameter for a constant Stand Density Index, SDI = 925 trees/ha, (regular staggered forest, $v_0 = 10$ m/s, snow properties: Case 2).

By the way, for practitioners it is properly more common to speak of number, N, of trees per hectare, instead of density.

Reply: We use the term forest density to be consistent with the referred studies (Teich et al., 2012 a, b, 2014; Feistl et al., 2012, 2014; Casteller et al., 2018; Perzl et al., 2021). We have further clarified the definition of the forest density as the number of trees per hectare in the revised manuscript.

Instead of using the ambiguous expression forest cover, it is the basal surface area per hectare that is meant here.

Reply: Revised.

Otherwise, the paper is well written and is a valuable contribution to an important question in respect to avalanche hazards and its mitigation.

Reply: We thank the reviewer for this positive comment.

References:

- Abegg, M., Brändli, U.-B., Cioldi, F., Fischer, C., Herold, A., Meile, R., Rösler, E., Speich, S., and Traub, B.: Swiss national forest inventory- Result table No. 815491: forest area Birmensdorf, Swiss Federal Research Institute WSL, https://doi.org/10.1103/PhysRevLett.86.71,2020.
- Casteller, A.; Häfelfinger, T.; Donoso, E.C.; Podvin, K.; Kulakowski, D.; Bebi, P. Assessing the interaction between mountain forests and snow avalanches at Nevados de Chillan, Chile and its implications for ecosystem-based disaster risk reduction. Natural Hazards and Earth System Sciences.2018,18, 1173–1186, https://doi.org/10.5194/nhess-2017-348, 2018.
- Christen, M., Kowalski, J., and Bartelt, P.: RAMMS: Numerical simulation of dense snow avalanches in three-dimensional terrain, Cold Regions Science and Technology, 63, 1–14, https://doi.org/https://doi.org/10.1016/j.coldregions.2010.04.005, 2010.
- Feistl, T.: Vegetation Effects on Avalanche Dynamics, Ph.D. thesis, 2015.
- Feistl, T., Bebi, P., Bühler, Y., Christen, M., Teich, M., and Bartelt, P.: Stopping behavior of snow avalanches in forests, in: Proceedings of the International Snow Science Workshop ISSW. Anchorage, Alaska, pp. 420–426, 2012.
- Feistl, T., Bebi, P., Teich, M., Bühler, Y., Christen, M., Thuro, K., and Bartelt, P.: Observations and modeling of the braking effect of forests on small and medium avalanches, Journal of Glaciology, 60, 124–138, https://doi.org/10.3189/2014JoG13J055, 2014.
- Perzl, F., Bono, A., Garbarino, M., and Motta, R.: Protective Effects of Forests against Gravitational Natural Hazards, https://doi.org/10.5772/intechopen.99506, 2021.
- Reineke, L. H.: Perfection a stand-density index for even-aged forest, Journal of Agricultural Research, 46, 627–638, 1933.
- Teich, M., Bartelt, P., Grêt-Regamey, A., and Bebi, P.: Snow Avalanches in Forested Terrain: Influence of Forest Parameters, Topography, and Avalanche Characteristics on Runout Distance, Arctic, Antarctic, and Alpine Research, 44, 509 – 519, https://doi.org/10.1657/1938-4246-44.4.509, 2012a.
- Teich, M., Vasella, I., Bartelt, P., Bebi, P., Feistl, T., and Grêt, A.:Avalanche simulations in forested terrain: a framework towards a Bayesian probabilistic model calibration, In Proceedings, 2012 International Snow Science Workshop ISSW, Anchorage, Alaska, p. 632, 2012b.
- Teich, M., Fischer, J.-T., Feistl, T., Bebi, P., Christen, M., and Grêt-Regamey, A.: Computational snow avalanche simulation in forested terrain, Natural Hazards and Earth System Sciences, 14, 2233–2248, https://doi.org/10.5194/nhess-14-2233-2014, 2014.

---

## Author Comment (AC2)

**Response to Reviewer #2's comment on "Detrainment and braking of snow avalanches interacting with forests"**

Louis Védrine, Xingyue Li*, and Johan Gaume
*Correspondence: xingyueli@tongji.edu.cn
January 26th, 2022

We thank Referee #2 for his or her insightful comments and helpful advice, which help to improve the quality of our paper. The following provides our point-to-point responses to the general comments, specific comments, and technical corrections from the reviewer.

This manuscript discussed the snow avalanche protection efficiency of the forests with applying the 3D MPM combined with the modified cam Clay model. This approach developed by the authors' group looks very powerful and promising to break through various aspects of snow avalanches from its initiation to dynamics. In actual, 3D MPM provides overwhelming useful information comparing to the previous depth averaged model. Further, once the model has been established it can be applicable under the various conditions. Taking into conceivable factors, the deduced relations in this study will be quite useful for the practical applications, I believe. Although it is out of scope here, I hope the authors proceed the research on the other aspect as well: how the forest stabilizes the snowpack and prevent from the snow avalanche release.

All the methodologies and discussions are well organized and carefully described. So, I believe this article is worth publishing. However, I am glad if the authors can clarify and improve some of the points listed below before the publication.

Reply: We thank this reviewer for the encouraging comments, and we appreciate the reviewer's interest in the effect of forest on the stabilization of the snowpack, which is indeed very important especially for avalanche release. We have conducted preliminary simulations on this topic and will keep working on it. This has been added to the discussion of the manuscript.

In addition, we have further improved the manuscript thanks to the reviewer's comments below.

Line 47: K is used for not only the detrainment coefficient but the material bulk modulus in eq. (5).

Reply: We thank the referee for noticing this issue. In the revised manuscript, we use K for the detrainment coefficient only, and we replaced the bulk modulus by its expression: $\frac{E}{3(1-2\nu)}$ with $E$ the young's modulus, $\nu$ the Poisson's ratio.

Please check fonts of variables. They are not always unified in the text, the figure and figure captions: italic or vertical, such as spacing of trees "e".

Reply: Checked and corrected.

Line 102: "CFL" means Courant-Friedrichs-Lewy Condition? Since it is not so common outside the numerical simulation field, the brief explanation is preferable, if you dare to use.

Reply: Yes. We have added more details in the revision.

Line 121: M=0.8 snow, defined as warm shear regime, agreed well with the field observations. On the other hand, Case 3 of M=1.2 is also set as the warmer case in Table 2. So, the Case 2 of M=0.8 snow correspond to the snow which is dry but the temperature is close to zero? I

am curious whether you have got the information of snow properties by Feistl et al.? Are these reasonably agree with the physical parameters given in this study? In this study snow properties are designated with snow friction coefficient: M, the tension compression ratio: β, the hardening factor ζ, and initial consolidation pressure p0ini. Although the general features of snow defined with these parameters are explained briefly, it will be glad if the authors clarify the relations more specifically with the snow properties we usually observe in the field.

Reply: The snow heights behind the trees given by Feislt et al. (2014) are obtained from observations of 7 different avalanches (dry to wet). Their illustration shows some snow granules (Fig. 2e in the manuscript), which can correspond to snow of intermediate cohesion and friction. The snow properties in Case 2 are calibrated with the data reported by Feistl et al. (2014). Therefore, the snow in case 2 with M = 0.8 can be regarded as moist snow. Based on the calibrated snow properties in Case 2, we have further modified the snow friction and cohesion to get a relatively colder snow in Case 1 and warmer snow in Case 3. This has been clarified in the manuscript.

We recognize that due to the scarcity of snow triaxial tests (Desrues et al.1980; Scapozza and Bartelt 2003), it is difficult to calibrate different types of snow modelled with critical state soil mechanics as in this study. However, the snow properties in our model do have their physical basis and can be determined according to physical properties of snow and/or parametric study (Li et al., 2020). In particular, the snow friction parameter M can be linked to the internal friction of snow as $\phi = sin^{-1}\left(\frac{6M}{3+M}\right)$ (Sadrekarimi and Olson, 2011). The cohesion parameter β is the ratio between the tensile and compressive strength. Given the compressive strength as $p_0$, the tensile strength of snow can be obtained as $\beta p_0$. The hardening factor $\xi$ reflects how fast the load increases with the deformation in the plastic stage. We have further clarified the snow properties in the revised manuscript.

According to our systematic study on the effect of snow properties on the avalanche behavior (Li et al., 2020), it was found that the tensile strength $\beta p_0^{ini}$ and $\beta M$ consistently increase from cold to warm avalanches, which suggests that these two terms control the different snow behaviors. By choosing Case 2 as a reference case, $\beta p_0^{ini}$ =9 kPa, $\beta M$= 0.24, we study the influence of the snow properties by modifying this pair of parameters. Firstly, to reproduce the behavior of a dry snow, we decrease the tensile strength $\beta p_0^{ini}$ =0 kPa and $\beta M$=0, this is the case 1 of our study, which corresponds to a cohesionless and low friction snow. And to reproduce the behavior of a wet and warm snow, we increase the tensile strength $\beta p_0^{ini}$=9 kPa and $\beta M$=0.36, this is the case 3 of our study, which corresponds to a relatively cohesive and frictional snow.

Further, do you have any idea how you can validate that the model is also appliable for the dry snow?

Reply: The snow properties of case 1 give the behavior of a dry cohesionless granular flow, whose free surface is continuous. It would be interesting in the future to compare this model with mass balances in the forest for a cold snow avalanche.

Table 2: The hardening factor ζ is set the same for both dry and wet snow. Is this reasonable?

Reply: We chose the same hardening factor $\xi$ because there is currently not enough experimental data for its calibration for different types of snow. This parameter characterizes the rate of the hardening and its effect on the snow behavior depends on other parameters $M, \beta, p_0^{ini}$. We checked that this parameter does not affect the presented results but would affect the densification process (especially behind the trees). In the future, model parameters could be evaluated based on extensive triaxial tests for different snow types.

Figure 2: No snow depositions are found in (b) after the rectangular domain. Does it mean all the snow went through?

Reply: The view for the previous figures 2a&b&c was a side view of the wedge formed behind a group of 3 trees, the snow particles on the surface of the wedges are moving which leads to the confusion. We have changed to a view of a cross section passing through the center of the middle tree in order to have a better visualization of the shape of the formed wedges (Fig.R1).

[Figure]

Figure R1. Cross-sectional view at the center of the middle tree of the snow stopped behind a group of three trees for different values of M: (a) M=1.2, (b) M=1, (c) M=0.8; accumulated snow behind a group of tree trees with a diameter of 1 m: (d) warm shear regime M=0.8, (e) observations (Feistl et al., 2014).

Table 2: Density of Case 1 which is explained as cold "dense" regime is still 200 kg/m3?

It is the same as other two warmer cases? I do have an impression that the density of the dry avalanche is usually lighter than the wet one. In particular, densities of granules and blocks in Case 3 are still 200 kg/m3? I wonder densities described here perhaps show bulk densities including air and snow?

Reply: Although the initial snow density is fixed as 200kg/m$^3$ for all the cases (Li et al., 2020, 2021), the avalanche density changes during the flowing process according to the properties of the snow and the hardening law. It has been further clarified that the 200kg/m$^3$ is the initial density of the snow.

In addition, as the adopted material point method is a continuous approach and each material point corresponds to a piece of snow instead of a real snow particle, the density is therefore the bulk density including air and snow. It has been further clarified in the revised manuscript that the density is the bulk density of snow.

Line 185: Although it does not matter much, I am a bit worried that authors use the word "impact" frequently. Since this article deals with the avalanche impact on the forest in actual, I prefer rewording such as "action", "affect" and "influence" to avoid the misleading.

Reply: Thanks for the reminder. We have replaced the word "impact" with "affect" and "influence" in the revised manuscript.

Figure 6: Eq. 3.5 and 3.4 in the figure must be Eq. 8 and 9.

Reply: Revised.

Is it just a coincidence that Case 1 maximum and Case 2 final agrees quite well? Further, please specify the reason why Case 3 is not shown here.

Reply:

i) We thank the reviewer for this remark, we have further clarified that this is a coincidence to avoid the confusion.

ii) Initially, Case 3 was not shown here because it gives a similar trend as cases 1 and 2. In addition, we wanted to highlight the comparison between cases 1 and 2 and the model proposed by Feistl et al. (2014) (Eq. 8 in the manuscript). However, recognizing that this may raise concerns from the reader, we have added case 3 in the revision.

The case 3 due to a high internal friction and cohesion, stopped the entire avalanche for large tree diameter, thus a saturation effect is observed, especially for the points with a diameter of 1.2 and 1.4m. This point has also been clarified in the caption of Figure 6 of the revised manuscript, as shown in Figure R2.

[Figure]

Figure R2. Evolution of the detrainment mass observed for the one tree arrangement and the fits with the tree diameter for 3 flow regimes: Case 1, Case 2, and Case 3 with respectively a front velocity of 12.5 m/s,10.9 m/s and 10.75 m/s, 'maximum' refer to the maximum mass stored behind the tree and 'final' refer to the final mass stored. A slope of 30°, and a top wedge angle of 60° (from measurements Feistl et al. (2014)) are used for the theoretical model (Eq. 8). The removed point denotes a special case not considered in the proposed square relation in Eq. 9, since the entire avalanche in this case is stopped due to the low flow velocity and the high tree diameter. * please note that it is a coincidence that Case 1 maximum and Case 2 final agree well.

Figure 8: Again, here why case 1 is not introduced?

Reply: Similarly, as the previous comment, we initially did not add case 1 as it gives a similar trend as cases 2 and 3. To avoid the confusion, we have added case 1 with an initial speed $v_0 = 10$ m/s in the revised manuscript.

Figure 9: Please explain physically why the detrainment mass is proportional to the third power of the forest density.

Reply: Assuming no interaction between trees, increasing the number of trees would lead to a linear relationship with the mass stopped. However, a second phenomenon is added, there is a collective effect of the trees on the stopped mass. Indeed, the increase in forest density leads to the reduction of the spacing between the trees, which leads to a higher snow jamming and the formation of snow arches between the trees. In addition, this results in a higher gradient of speed between trees which leads to a higher dissipation by friction. The above explanation is in terms of the physical process but does not directly give us the third power law, which corresponds to the best fitting model based on the obtained simulation data. This has been clarified in the manuscript.

Table 3: According to Figure 8, the relation between the detrainment mass and tree diameter for Case 2 and Case 3 are obviously different even though the front velocities are almost the same (11.2 m/s and 10 m/s). Thus, I do not understand the reason why you can use the same p3 and p4 for all three types of snow as shown in Table 3. Further, although no results are not introduced in 4.2.2, I am wondering the snow type does not give any effect on the forest density consequence.

Reply: Indeed, with different snow types, the detrainment mass will be different. However, as introduced in section 4.1.2, it is the parameters $p_1$ and $p_2$ that account for the different snow properties, instead of $p_3$ and $p_4$. In Table 3, the parameters are for cases with different snow properties, therefore, $p_1$ and $p_2$ have different values while $p_3$ and $p_4$ are identical for the three cases.

To further explain the same $p_3$ and $p_4$ for the three cases in Fig. 8, we normalize the detrainment mass obtained from Fig. 8 (or Eq. 9 in the manuscript) by the detrainment mass with different snow properties and front velocity (Eq. 7 in the manuscript), we obtain a unique curve as shown in Fig. R3, and we identify the same p3 and p4 for all the cases.

[Figure]

Figure R3. Evolution of the detrainment mass normalized with the front velocity and the snow properties (Eq. 7) for different tree diameters (regular staggered forest, $e = 8$ m).

Line 273: Please explain why the detrainment energy, when the avalanche reaches the bottom of the valley, is similar between the random and the regular staggered arrangements, in spite of the fact that the final stopped mass with both arrangements are largely different.

Reply: In the case of regular configurations, due to their geometric arrangement, very large deflections are observed. This results in a significant lateral spreading and the delay (or velocity reduction) of part of the avalanche due to numerous collisions (please see supplementary movie 5). Contrary to the random case, this snow is not stopped, but contributes to reducing the velocity/severity of the avalanche. The study of the detrainment energy enables us to both consider the stopped mass and the mass with reduced velocity.

Line 281: Random arrangements is the most effective to stop the avalanche. I am curious how you suggest to people in charge who take care of and manage the forest? Do you say the forest should leave as it is without artificial maintenance? Foresting should be done randomly?

Reply: Indeed, this study shows that in this idealized condition, the random arrangement is the most effective to stop the avalanche. Random foresting can therefore be part of the risk mitigation measures in high-risk area. Another important point of this study is that for planted forests, the staggered arrangement is clearly more effective than the aligned forests. The staggered arrangement can be easily achieved with current devices and without disrupting the forestry sector, to significantly increase the protective effect of forests. However, this study is carried out under idealized conditions and focuses only on the avalanche dynamics. Therefore, further studies are needed, on investigating the effect of the forest arrangement on the release of snow avalanches.

References:

- Bartelt, P. and Stöckli, V.: The influence of tree and branch fracture, overturning and debris entrainment on snow avalanche flow, Annals of Glaciology, 32, 209–216, https://doi.org/10.3189/172756401781819544, 2001.
- Desrues, J., Darve, F., Flavigny, E., Navarre, J., & Taillefer, A.: An Incremental Formulation of Constitutive Equations for Deposited Snow. Journal of Glaciology, 25(92), 289-307. doi:10.3189/S0022143000010509,1980.
- Feistl, T., Bebi, P., Teich, M., Bühler, Y., Christen, M., Thuro, K., and Bartelt, P.: Observations and modeling of the braking effect of forests on small and medium avalanches, Journal of Glaciology, 60, 124–138, https://doi.org/10.3189/2014JoG13J055, 2014.
- Li, X., Sovilla, B., Jiang, C., and Gaume, J.: The mechanical origin of snow avalanche dynamics and flow regime transitions, The Cryosphere, 14, 3381–3398, https://doi.org/10.5194/tc-14-3381-2020, 2020.
- Li, X., Sovilla, B., Jiang, C., and Gaume, J.: Three-dimensional and real-scale modeling of flow regimes in dense snow avalanches, Landslides, https://doi.org/10.1007/s10346-021-01692-8, 2021.
- Sadrekarimi, A. and Olson, S.: Critical state friction angle of sands, Géotechnique, 61, 771–783, https://doi.org/10.1680/geot.9.P.090, 2011.
- Scapozza, Carlo & Bartelt, Perry. Triaxial tests on snow at low strain rate. Part II. Constitutive behaviour. Journal of Glaciology. 49. 91-101. 10.3189/172756503781830890, 2003.

---

## Author Comment (AC3)

**Response to Reviewer #3's comment on "Detrainment and braking of snow avalanches interacting with forests"**

Louis Védrine, Xingyue Li*, and Johan Gaume
*Correspondence: xingyueli@tongji.edu.cn
January 26th, 2022

We thank Referee #3 for his or her detailed comments and valuable suggestions, which helped us to improve the quality of the paper. Our point-to-point replies to the comments of the reviewer are summarized below.

The manuscript "Detrainment and braking of snow avalanches interacting with forests" by Vedrine et. Al. is a computational study on how detailed numerical modelling approaches can contribute to investigate how gravitational mass flows interact with obstacles. The forest (obstacles) can offer a protective effect which reduces the size or frequency of avalanches by stopping the formation of avalanches or reducing the magnitude of an event. This work focuses on quantifying the mass and energy reduction capabilities of forest (detrainment and braking) in the transit zone of a small or medium sized avalanches by detrainment, which reduces the kinetic energy of the avalanche by removing mass.

The work highlights the possibility of using purely numerical methods (MPM) to quantify the potential effect of forests and parameterize the forest snow interaction within simple relationships of terrain (slope), flow (velocity) and forest parameters (density). The simulation experiments were carried out on a generic slope with a constant slope angle, examining the influence of mainly avalanche type/velocity and different forest formations/density with respect to mass/energy reduction. The parameters of the MPM avalanche model are defined by prior experiments to resemble the behavior of colder to warmer flow regimes and calibrated with regards to snow forest interactions based on a single documented field observation. The single event validation could be considered as weakness of the study. However it is known that corresponding data is sparse – but it would be interesting to comment on other the possibility of other parameter combinations that might lead to similar results, or how they would change for another observation example. Another clarification would be desirable for the definition (and numerical implementation) of detrained mass (see specific comments below) in the MPM model and how changing boundary conditions (slope angle) would influence the results.

Reply:

i) Indeed, as underlined by Stritih (2021), there are scarce empirical data on the braking effect on avalanche, especially at local scale. However, as suggested by reviewer 1, the revised manuscript have further considered the study of the stand density index (Reineke, 1933) which is a widely used parameter by practitioners. This parameter reflects a combined effect of tree diameter and forest density, the law (Eq.11 in the manuscript) can be used by substituting one of these two parameters with the stand index density.

ii) A clarification of the definition of the detrainment mass has been made in the revision.

iii) The effect of the slope angle is indeed important. The first order effect of the slope angle is on the front velocity, as shown by Li et al. (2020). In fact, the slope angle between the release zone and the forest has a direct effect on the front velocity which is quantified in equations 7 &11.

The second order effect is on the geometry of the snow wedges formed behind the trees. To study the effect of the slope angle, the one tree configuration (Fig. 1a in the manuscript) is

used with the release zone very close to the tree to reduce the effect of the front velocity. In addition, to exclude the effect of the velocity on the mass stopped, the mass has been normalized with a front speed of 10m/s (Eq.7 in the manuscript). As expected, the mass stopped decreases with an increasing slope angle (Fig. R1) as with the model proposed by Feistl et al. (2014). (Eq. 8 in the manuscript). However, our results in Fig. 7 suggest that the bed friction is a key parameter on the evolution of the detrainment mass, and that the dynamic of the avalanche needs to be considered. Nevertheless, as the effect of slope angle is closely related to other factors like avalanche velocity and bed friction, it is very difficult to quantify its individual effect, and we did not propose a law with slope angle in our study.

[Figure]

Figure R1. Evolution of the stopped mass with the slope angle for a bed friction of 0.5.

For future studies, we propose to calibrate the model for a given pair of parameters (bed friction and slope angle). Indeed, we have shown that it is possible to find a constant $\alpha$, that enables us to use the law Eq.11 for different slope angles such as $M_d(\theta = 35°) = \alpha M_d(\theta = 30°)$. This parameter $\alpha$ accounts for the geometry change of the wedges. As illustrated in Fig. R2, the evolution trend of the detrainment mass with the different slope angles does not change, showing the validity of the proposed Eq. 11. For this specific case, the value of $\alpha$ is 1/1.7.

[Figure]

Figure R2. Evolution of the detrainment mass with the tree diameter for 2 different slope angles. For the model used in the case of a slope at 35 °, we use a geometric correction factor of $\alpha = 1/1.17$.

Generally, the manuscript is well written and well organized, providing suitable figures and supplementary material. Some possible enhancements include the figures in the energy analysis and the consistency between equations and figures (e.g. fig 5, "velocity" = "v_f" in eq. 7, more examples in the specific comments (e. g. Fig 10)).

Reply: We thank the reviewer for the helpful comments, according to which we have revised the manuscript.

**Specific comments :**

- l 11: "wet compared to dry snow": Since this is a numerical study i would suggest to rephrase (or is there evidence in field observations?): "for the parametrizations of cold to warm snow"

Reply: We have rephrased the description as suggested, since there is indeed no direct field data or experimental data of all the snow properties used in this study. The snow properties in Case 2 are calibrated with the data reported by Feistl et al. (2014). Based on Case 2 and our parametric study (Li et al., 2020), we have further modified the snow friction and cohesion to get a relatively colder snow in Case 1 and warmer snow in Case 3.

In addition, we have further clarified the physical meaning of the snow properties adopted in this study as detailed in the reply to reviewer 2.

- l 36: What about the study of "Brožová, N., Fischer, J. T., Bühler, Y., Bartelt, P., & Bebi, P. (2020). Determining forest parameters for avalanche simulation using remote sensing data. Cold Regions Science and Technology, 172, 102976 (11 pp.). https://doi.org/10.1016/j.coldregions.2019.102976". Does it relate or include relevant data to evaluate the results of this study?

Reply: This study is based on and complementary to the work of Feistl (2015) where a value of the detrainment coefficient K for different types of forest is proposed. The study by Brožová et al. (2020) assesses the quality of forest structural parameters obtained from remote sensing data using two different methods and using the value of K given by Feistl (2015). They compare the effect of the forest parameter on the avalanche flow.

This study essentially aims to make the method proposed by Feistl et al. (2014) operational by facilitating the determination of forest parameters and no new comparison with Feistl (2015) is provided in terms of the braking of forest on avalanches.

We have added discussion on the study by Brožová et al. (2020) in the revision, because the method proposed by Brožová et al. (2020) is an essential step in the process of risk prevention, for example within the framework of bayessian networks (Stritih, 2021). Indeed, remote sensing can help to identify the forest parameters used in this study, which allows a simple application, and gives access to information which was not considered in the study but would be interesting to be considered in the future such as surface roughness.

- Table 2 "Case 1..3": I think it could be beneficial to name the cases "cold/intermediate/warm" here and throughout the paper to make it easier for the reader (and please check consistency of warm/wet and cold/dry throughout the paper).

Reply: As discussed above, since the snow parameters are not directly linked to the field observations and rather represent the mechanical properties which can correspond to a cold, intermediate or warm snow, we keep the name cases 1-2-3. This has been clarified in the revised manuscript.

- l 143, "some arches appear in case 3": Can you comment on how "arches" (surges?) are defined in this context?

Reply: In this context, an arch is formed by stopped snow between two trees due to the jamming effect (Feislt et al., 2014). This phenomenon occurs when the size of the wedges behind the trees is so large that two wedges intersect. This phenomenon is more obvious when the snow is highly cohesive and frictional and when the spacing between the trees is small. This arch phenomenon is similar to the jamming of a granular flow in a two-dimensional hopper (Lai et al., 2001). Further clarification has been made in the revised manuscript.

- l 165 "detainment mass": Does this mean snow is considered detrained if its velocity<0.5m/s and adds to "M_stopped" – how does it relate to (frictional) stopping – are these numerical of flow model quantities? A clarification on the definition of "M_stopped" seems to be crucial for the paper and could be included at this point, particularly the difference between "stopped" (Fig 5), "maximum" and "final" (Figs 5,6) or "stored" (Fig 7). Please also check the corresponding units [kg] or [kg/m^2] used for "M_stopped".

Reply: We thank the reviewer for pointing out the ambiguous notations. We define the detrainment mass as the total mass of the snow particles which have a velocity smaller than 0.5 m/s during the flowing process. To be in line with the notation used by Feistl et al. (2014) we use the notation $M_d$ for the detrainment mass in (kg/m²) (per unit of area).

When the stopped mass of a single tree (e.g. Fig. 6) is of interest, we use the mass stopped $m_d$ (kg) in the revision. In this case, we do not normalize the stopped mass with the area around the tree as the area is difficult to define. Please note that in Fig. 7, although there are multiple trees, it is the stopped mass by individual trees that we compare to Fig. 6, therefore, $m_d$ (kg) is also used in Fig. 7 in the manuscript.

The mass stopped behind the trees evolves with time due to the changing shape of the wedges over time. Therefore, we study two critical quantities (Fig. 6), the maximal mass stopped which refers to the maximum mass stopped behind the tree over time, and the final mass stopped which refers to the mass stopped at the end of the simulation. We choose to plot these two quantities because the maximum mass stopped is related to the global detrainment mass over the entire forest and the final mass stopped corresponds to the mass of snow which is observed in the field measurement.

- l 192: Does maximum detrained mass refer to the maximum over time (detrained snow = v> 05m/s?, see comment above)?

Reply: Yes, please see the detailed reply above.

- l 214: Should this not be p_3 and p_4?

Reply: Revised.

- l 243: This sentence is confusing, please clarify: decreases linearly, as function of ..?

Reply: The sentence has been revised as follows.

In addition, in terms of potential energy, whereas the potential energy without forest decreases linearly as a function of time, with forest, due to the mass which stays on the slope, the potential energy decreases more slowly with time. Consequently, at the end of the simulation, the potential energy does not vanish due to the detrainment mass.

- l 281: Is the statement that the random distribution has a higher protective effect valid after checking one specific distribution?

Reply: Thanks for the comment. To validate the result obtained on the small slope (Fig. 13 in the manuscript), we carried out the energy study on a larger scale with another random forest as shown in Fig. R3, with a forest length of 160 m instead of the original 40 m. We observe a stopped mass of 7735 kg and 20540 kg for respectively the staggered and the random arrangement, a similar dissipation for the two cases and a slightly higher detrainment energy for the regular arrangement (Table R1). These conclusions are in agreement with those obtained from the original study at a smaller scale, the detrainment energy is similar between the two arrangements, but the mass stopped, and the runout-distance (Fig. R3) suggest that the random arrangement has a higher protective effect.

[Figure]

(a)  (b)  Velocity (m/s)
0  3  6  9  12

Figure R3. Flow profile for 2 different forest arrangements: (a) regular staggered, (b) random, at t = 33 s, snow type: Case 2.

| Forest arrangement | $m_d$(kg) | $E_{pd}$(MJ) | $\tilde{E}_f$ (MJ) |
|---|---|---|---|
| Regular staggered | 7735 | 96.3 | 6.0 |
| Random | 20540 | 88.6 | 6.1 |

Table R1. Mass stopped, detrainment and dissipation energy for two types of forest arrangement.

- l 301: Please double check the argument with low/high velocities.

Reply: It has been double checked that the argument is true, but the description of the velocities is unclear. Therefore, we have revised the sentence as follows: When the avalanche velocity is high, the plateau stage of the detrainment mass obtained from our study is higher than the decreasing stage by Feistl et al. (2014). Consequently, an implementation of our

proposed velocity-based model would lead to higher detrainment mass, which results in lower runout distance and smaller velocity.

**Equations:**

- Why is "." Sometimes used as mathematical symbol for multiplication in the manuscript (e.g. eq. 7-11)?

Reply: The symbol has been removed to avoid the confusion.

**Figures:**

- Fig 2: "d" and "e" are used twice once for tree diameter and spacing and once for the forest arrangements. "snow profile" should rather be "vertical velocity profile"?

Reply: The tree diameter "d" and spacing "e" have been written in italic. The snow profile has been changed to "cross-sectional view at the center of the middle tree of the snow stopped" to be clear.

- Fig 6: legend wrong? 3.5 and 3.4 are the ones in Feistl et al. And eq. 3.4 is the same as eq.8?

Reply: Corrected.

- Fig 7: is this for case 2?

Reply: Yes, this has been clarified in the revised manuscript.

- Fig 10: is "M_d" from MPM the same as "M_stopped" in the previous figures (same, "M_d" from eq. 11 = "m_d"). What kind of r^2 is used? Is it possible to comment on how the different cases (1-2) are distributed in this figure?

Reply:

i) We have further revised and clarified the adopted notation in the manuscript. Please see detailed the reply to the previous comment on the notation.

ii) We use $r^2$ as the coefficient of determination which is equal to the square of the Pearson correlation coefficient.

iii) It is possible to differentiate cases 1-3 by using different markers as shown in Fig. R4. We notice that case 1 gives the lowest detrainment mass, and case 3 has the highest detrainment mass as expected. The original Fig. 10 has been replaced with this new figure in the revision.

[Figure]

Figure R4. Evolution of the detrainment mass predicted with the model (Eq. 11) and with the observation.

- Fig 11: Do both avalanches reach the bottom after 9 s (effect of forest on front velocity)?

Reply: Yes, with and without forest, the front of the avalanche reaches the bottom of the slope at the same time because there is always a part of the front that is not deviated and that crosses the forest without any collision (please see supplementary movie 5 or Fig. R5).

We define the time when the avalanche reaches the bottom of the slope when the first 1% of the front reaches the bottom of the slope. We have chosen a criterion based on a small percentage of the avalanche front because in some cases a large part of the avalanche is stopped in the forest and will never reach the bottom of the slope, and in this case, we can assume that with and without forest the front of the avalanche reaches the bottom of the slope at the same time. But in fact, the decrease of the front velocity depends on the percentage of the avalanche that is considered as the front, and this decrease will be observed if we consider more particles as the avalanche front. This has been clarified in the revised manuscript.

[Figure]

Figure R5. Flow profile: (a) without forest and (b) with a regular staggered forest (Case 2, $v_0$=6 m/s, t=8.75 s).

References:

- Brožová, N., Fischer, J.-T., Bühler, Y., Bartelt, P., and Bebi, P.: Determining Forest parameters for avalanche simulation using remote sensing data, Cold Regions Science

and Technology, 172, 102 976, https://doi.org/https://doi.org/10.1016/j.coldregions.2019.102976, 2020.

- Feistl, T.: Vegetation Effects on Avalanche Dynamics, Ph.D. thesis, 2015.
- Feistl, T., Bebi, P., Teich, M., Bühler, Y., Christen, M., Thuro, K., and Bartelt, P.: Observations and modeling of the braking effect of forests on small and medium avalanches, Journal of Glaciology, 60, 124–138, https://doi.org/10.3189/2014JoG13J055, 2014.
- Lai, P.-Y., Pak, H., and To, K.: Jamming of Granular Flow in a Two-Dimensional Hopper, Physical Review Letters, 86, https://doi.org/10.1103/PhysRevLett.86.71, 2001.
- Li, X., Sovilla, B., Jiang, C., and Gaume, J.: The mechanical origin of snow avalanche dynamics and flow regime transitions, The Cryosphere, 14, 3381–3398, https://doi.org/10.5194/tc-14-3381-2020, 2020.
- Reineke, L. H.: Perfection a stand-density index for even-aged forest, Journal of Agricultural Research, 46, 627–638, 1933.
- Stritih, A.: Dealing with Uncertainties in the Assessment of the Avalanche Protective Effects of Forests, https://doi.org/10.5772/intechopen.99515, 2021.